# Image quality improvement of liver ultrasound using unsupervised deep learning

Jaeyoung Huh[1]◉, Joo Hyeok Choi[2,3]◉, Eun Sun Lee[2,3]*‡, Jong Chul Ye[1,4]*‡, Jeong Eun Lee[5], Hyun Jeong Park[2,3], Byung Ihn Choi[2]

**1** Department of Bio and Brain Engineering, Korea Advanced Institute of Science and Technology (KAIST), Daejeon, South Korea, **2** Department of Radiology, Chung-Ang University Hospital, Seoul, South Korea, **3** Chung-Ang University, College of Medicine, Seoul, South Korea, **4** Graduate School of AI, Korea Advanced Institute of Science and Technology (KAIST), Daejeon, South Korea, **5** Department of Radiology, Chungnam National University Hospital, Chungnam National University College of Medicine, Daejeon, South Korea

◉ These authors contributed equally as first authors to this work.
‡ These authors contributed equally as co-corresponding authors to this work.
* seraph377@gmail.com (ESL); jong.ye@kaist.ac.kr (JCY)

## Abstract

Chronic liver disease (CLD) and subsequent liver cirrhosis (LC) are common causes of death and healthcare-related socio-economical costs worldwide. Ultrasound (US) is the first-line imaging modality for assessing the liver and associated hepatocellular carcinomas. Poor quality liver US images caused by aging or inadequate management of US equipment, can pose significant challenges in both diagnosis and treatment. From this perspective, the aim of this study was to enhance and assess the image quality of liver US obtained from an older, lower-performing device using a deep learning approach. A neural network based on a switchable cycle generative adversarial network (CycleGAN) was trained in an unsupervised learning setting, with low-quality images as inputs and high-quality images as targets. The study included consecutively acquired grey-scale liver US examinations from both a 12-year-old and a 4-year-old US device. Images from the older device served as inputs, while images from the newer device were used as targets for the deep learning-based algorithm. Image quality was evaluated by two experienced reviewers. The algorithm significantly improved the brightness, contrast, and overall quality of the reconstructed liver US images (p < 0.001), as assessed by both reviewers. However, no significant differences in image resolution and reverberation artifacts were noted by one of the reviewers. The weighted kappa values for image quality and diagnostic performance ranged from 0.225 to 0.838, indicating fair to almost-perfect inter-reader agreement. The proposed algorithm effectively enhances low-quality liver US images to high diagnostic quality, thereby potentially supporting clinical assessment and intervention in patients with LC.

**Data availability statement:** All relevant data are within the manuscript and available from the DOI: 10.6084/m9.figshare.27011980 URL: https://figshare.com/articles/dataset/Image_quality_improvement_of_liver_ultrasound_using_unsupervised_deep_learning/27011980.

**Funding:** The author(s) received no specific funding for this work.

**Competing interests:** The authors have declared that no competing interests exist.

**Abbreviations:** ACG, Adaptive instance normalization code generator; AdaIN, Adaptive instance normalization; AUC, receiver operating curve; CLD, chronic liver disease; CNR, contrast noise ratio; CR, contrast ratio; CT, computed tomography; CycleGAN, cycle generative adversarial network; FID, Frechet Inception Distance; GB, gallbladder; GLCM, Grey-Level Co-occurrence Matrix; HBV, hepatitis B virus; HCV, hepatitis C virus; I2I, image-to-image; LC, liver cirrhosis; MRI, magnetic resonance imaging; SSIM, Structural Similarity Index Measure; US, ultrasound.

## Introduction

CLD and LC remain major causes of mortality and healthcare burden worldwide [1–3]. Despite advances in vaccination and antiviral therapy, the prevalence of CLD continues to increase, and the incidence of hepatocellular carcinoma has recently plateaued [4,5].

US is the first-line imaging modality for liver assessment and surveillance because of its accessibility and cost-effectiveness [6–8]. However, liver US image quality varies depending on equipment performance and maintenance status. Aging or inadequately maintained systems often produce images with low contrast and prominent speckle noise, which can affect diagnostic confidence. This issue is particularly relevant in local clinics and resource-limited settings where replacement of equipment or access to Computed Tomography (CT) or Magnetic Resonance Imaging (MRI) is limited, highlighting the need for practical image-enhancement solutions [9].

Recent advances in deep learning have enabled significant improvements in medical image analysis and enhancement across multiple modalities [10–14]. Nevertheless, most existing studies have focused on diagnostic classification or general US processing rather than directly restoring liver US images acquired from aging or low-performance systems.

Therefore, this study aimed to enhance liver US images obtained using older equipment through an unsupervised deep learning framework based on CycleGAN and to evaluate its effectiveness using both quantitative image metrics and clinician-based assessment [15].

This study retrospectively evaluated stored US images acquired between 2016 and 2018; real-time enhancement during scanning was not assessed and remains a direction for future development.

### Background

**Traditional and deep learning approaches for ultrasound image.** Before the adoption of deep learning, various filtering and model-based approaches were developed to improve US image quality, including adaptive shock filtering, bilateral filtering, and spatial-frequency–based smoothing [16–18]. Although computationally efficient, these methods often struggled to suppress speckle noise while preserving fine anatomical structures required for diagnosis.

Early machine-learning approaches relied on handcrafted features and conventional classifiers for liver disease assessment [19,20]. With the emergence of deep learning, convolutional neural networks and transfer-learning strategies substantially improved liver US analysis, including lesion detection, fatty liver quantification, and fibrosis staging [21–27].

Recent research has also explored US image quality enhancement using both RF-domain reconstruction and image-domain deep learning methods [28–31]. However, most prior work has focused on diagnostic tasks or general US enhancement rather than specifically improving liver US images acquired using aging equipment.

**Background of Image-to-Image translation.** Image-to-image (I2I) translation has become an active research area in deep learning for medical image enhancement. Conditional GAN-based frameworks such as Pix2Pix demonstrated

effective translation when paired data were available [32], while CycleGAN enabled unsupervised translation using unpaired dataset through cycle-consistency constraints [15]. Adaptive Instance Normalization (AdaIN) further enabled flexible style transfer by aligning feature statistics across domains [33].

Subsequent developments incorporated contrastive learning to improve unsupervised translation stability [34–36], and transformer-based architectures have been introduced to better capture long-range contextual information in medical image enhancement tasks [37]. Recent approaches such as StegoGAN have further improved structural preservation in non-bijective image translation scenarios [38].

These advances highlight the potential of unsupervised I2I translation for medical imaging applications, particularly in scenarios where paired dataset are unavailable, such as improving US image quality across different devices.

## Materials and methods

The institutional review board approved this retrospective study, and the requirement for informed consent was waived. (IRB number: 2112-012-19394).

### Dataset

From January to April 2022, we prepared two categories of datasets for training our deep learning-based image quality improvement algorithm, i.e., 1) liver US obtained by a US machine >10 years old, consequently having quality deterioration, served as an input, and 2) liver US obtained by a high-end US machine manufactured within last five years, served as a target as shown in Fig 1. Basically, all liver US scans included a minimum of 10 images (ranging from 10 to 38). These images encompassed suitable liver and gallbladder (GB) visuals, with no cine clips present. Additionally, all images were acquired using a 1–6 MHz convex transducer. For an input dataset, we randomly selected 500 liver US (training sets: validation sets: test sets = 350: 50:100) from 746 consecutively enrolled examinations from January 2016 to February 2018, performed by a hepatologist with 20 years of experience, using a 12-year-old US system (SSD-alpha 10 US

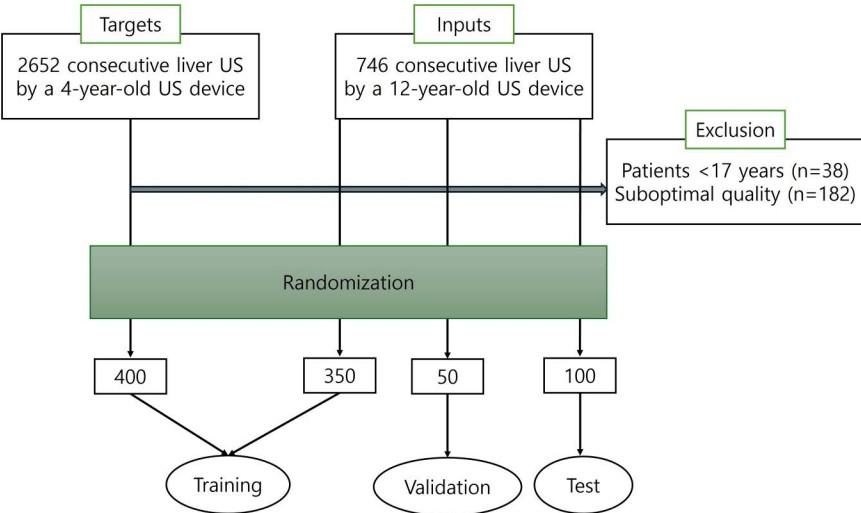

**Fig 1. Flowchart of patient and dataset selection for liver US image enhancement.** A total of 746 examinations obtained using a 12-year-old US system (input domain) and 2,652 examinations obtained using a newer high-end system (target domain) were screened. Exclusion criteria included patients younger than 17 years (n = 38), examinations predominantly consisting of color Doppler images (n = 3), and suboptimal studies according to the Korean Society of Ultrasound in Medicine guidelines (n = 179). After applying these criteria, eligible cases were randomly selected and divided into training, validation, and test sets. Images from the older device were used as the input dataset, and images from the newer device were used as the target dataset for unsupervised training.

System, Aloka Co., Ltd., Japan). For a target dataset, we randomly selected 400 out of 2,652 liver US performed by one of three board-certified abdominal radiologists with more than 15 years of experience (E.S.L., H.J.P., and B.I.C.) from December 2020 to December 2021. All US of target dataset were obtained by a high-end US machine (Aplio i900, Canon Medical Systems, Japan), manufactured within the last 5 years from the date of examination.

All US images used in this study were retrospectively retrieved from the institutional Picture Archiving and Communication System (PACS) in DICOM format. The analysis was performed on archived grey-scale B-mode images rather than raw radiofrequency (RF) data. Images were stored according to the institutional archival protocol, which applies consistent export settings across devices. The file format, bit depth, and compression policy were identical under the PACS storage system. Prior to training, all images were converted to a standardized resolution and normalized using identical preprocessing steps to minimize potential variability related to vendor-dependent export or compression differences.

We randomly selected all data for study periods; however, the following were excluded from the study population: 1) examinations for patients <17 years old (n = 38); 2) examinations that consisted of almost color Doppler images for liver transplantation (n = 3); and 3) suboptimal study according to the guidelines of the Korean Society of Ultrasound in Medicine, for liver US (n = 179) as shown in Fig 1 and Table 1. Regarding randomization of dataset, we used the function of random samples using Microsoft® Excel®.

## Switchable cycle generative adversarial network

CycleGAN is a representative unsupervised algorithm for I2I translation [15]. It contains two generators and discriminators trained in an adversarial manner. Specifically, a generator translates images from domain X to Y. Another generator then transforms images from domain Y back to the domain X.

Instead of using two generators, the recently proposed Switchable CycleGAN shown in Fig 2(a) utilizes a single generator with AdaIN code generator (ACG), so that its role for the forward and inverse transformation can be controlled by the AdaIN code. The AdaIN is the method for image translation by re-normalizing the generator feature-map using statistical information such as mean and variance [33]. The equation is as follows:

$$\text{AdaIN}(f, s) = \sigma(s) \left( \frac{f - \mu(f)}{\sigma(f)} \right) + \mu(s),$$

(1)

where $f$, $s$ denote feature map of input image and target style, respectively. The $\mu$, $\sigma$ represent mean and variance. Here, the $\mu(s)$, $\sigma(s)$ are generated from ACG and applied to each layer of the generator.

**Table 1. Table for dataset description.**

| Input dataset | |
|---|---|
| Included number | 746 |
| Randomly selected number | 500 |
| Training set: Validation set: Test set | 350: 50: 100 |
| **Target dataset** | |
| Included number | 2,652 |
| Randomly selected number | 400 |
| **Exclusion criteria** | |
| Examination for patients < 17 years old (n = 38) | |
| Examinations consisting primarily of color Doppler images (n = 3) | |
| Suboptimal studies according to the Korean Society of Ultrasound in Medicine guidelines (n = 179) | |

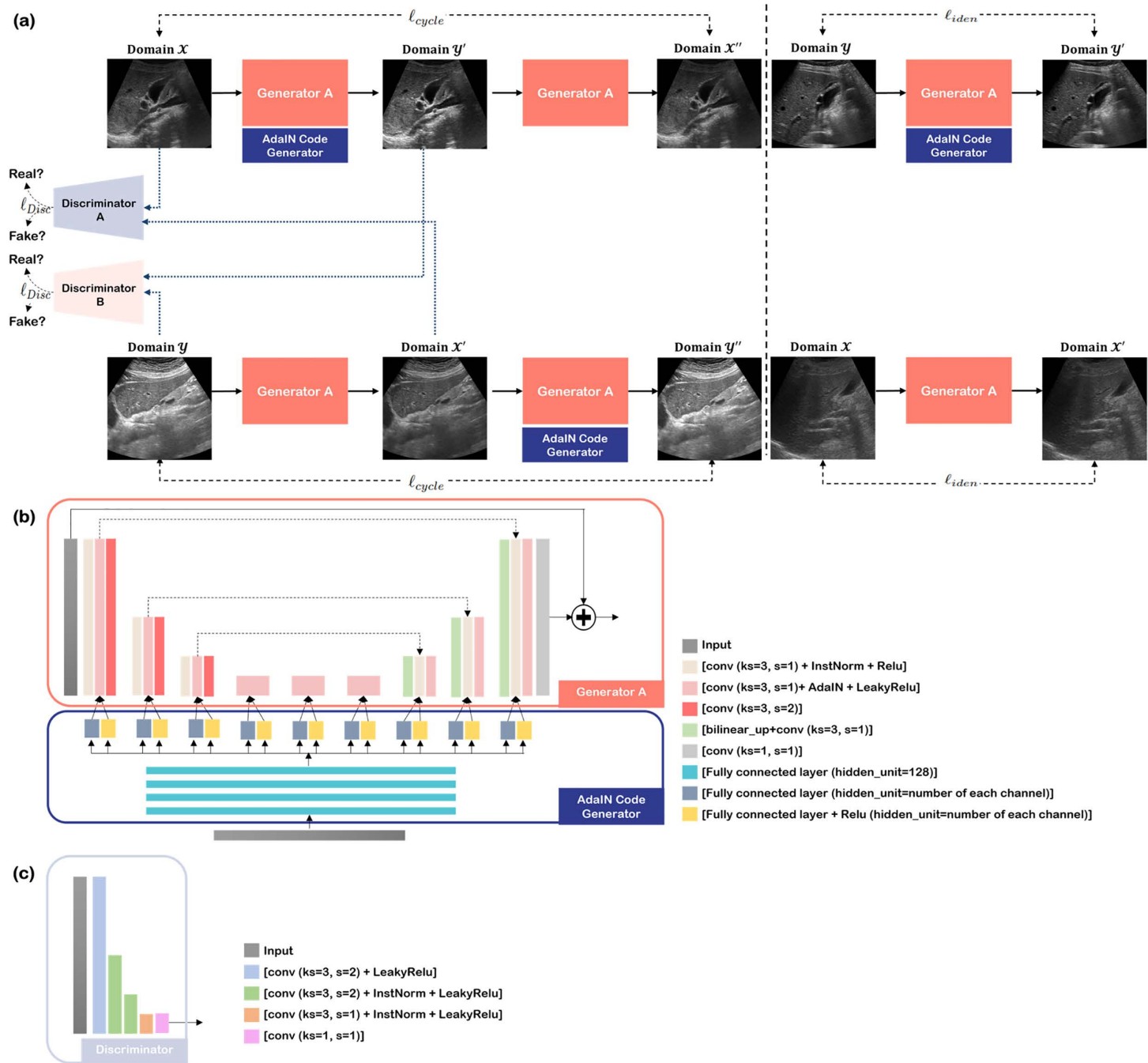

**Fig 2. Architecture of the proposed Switchable CycleGAN for liver US image enhancement. (a)** Overall framework of the Switchable CycleGAN. Images from the low-quality domain (Domain X) are translated to the high-quality domain (Domain Y) using a single shared generator (Generator A) modulated by an AdaIN code generated from the ACG. Cycle-consistency loss enforces reconstruction of the original domain, and identity loss constrains unnecessary modifications. Two discriminators (A and B) distinguish real and generated images in each domain. **(b)** Detailed architecture of the generator and ACG. The generator consists of convolutional, instance normalization, AdaIN-modulated layers, and up sampling blocks. The ACG produces channel-wise modulation parameters (mean and variance) that control domain translation within a shared feature space. **(c)** Architecture of the discriminator composed of convolutional layers with progressive down sampling to classify real versus synthetic images. *Note: CycleGAN, cycle generative adversarial network; US, ultrasound; AdaIN, adaptive instance normalization; ACG, AdaIN Code Generator.*

Compared with the conventional two-generator CycleGAN framework, the use of a single shared generator reduces model complexity and constrains domain translation within a unified feature space, which is advantageous for preserving anatomical structures in medical imaging tasks.

Like the conventional CycleGAN, our network is trained with two cycles. In the forward cycle, a low-quality US image in domain X is translated into a high-quality image in domain Y through a combination of the generator and ACG. It then returns to a low-quality image using only the generator so that it remains similar to the original input image. This condition is imposed as the cycle-consistency ($\ell_{\text{cycle}}$). The backward cycle proceeds as per the opposite translation procedure. While discriminator A is trained to distinguish the real from fake in domain X ($\ell_{\text{adv}}$), discriminator B determines the real and fake in domain Y. Additionally, the identity constraint was imposed for training stability ($\ell_{\text{iden}}$). The networks, such as generator, two discriminators, and ACG, were trained simultaneously. Other details are elucidated in the supplemental materials.

Since there are no ground-truth dataset, the proposed method was evaluated by visual inspection by two board-certified abdominal radiologists in two ways: One for image quality assessment and the other for diagnostic assessment.

## Computational performance and inference setting

The computational performance of the proposed model was evaluated during inference using a single NVIDIA RTX 3090 GPU. The model operates in a feed-forward manner without iterative optimization. For an input resolution of 256×256 pixels, the average inference time was approximately 30–40 ms per image, corresponding to approximately 25–33 frames per second (FPS) under single-image processing conditions.

## Assessment of imaging quality of liver US

Two board-certified abdominal radiologists with 15 years of experience (E.S.L. and J.E.L.) evaluated paired test dataset divided into original (n = 100) and post-processed (n = 100) sets randomly shuffled. Every liver US has multiple images, ranging from 10 to 35, covering the liver, GB, biliary tree, a part of the pancreas, and spleen under at least two planes. Image quality was evaluated using five categories, i.e., brightness, contrast, resolution, reverberation artifact, and overall quality, by a 5-scale scored system in Table 2. Images with optimal quality for accurate diagnosis are assigned a score of 5, while those with good quality for diagnosis are assigned a score of 4. Images with somewhat inadequate quality fall under score 3. Images with predominantly poor quality for diagnosis are categorized as score 2. Finally, images that do not meet diagnostic quality standards are assigned a score of 1.

## Diagnosis assessment

The two reviewers assessed the presence of LC, fatty liver, solid hepatic focal lesion detection, GB polyps, and gallstones for every data per patient, to compare the diagnostic performances between the original and post-processed sets. Among the patients, forty-seven patients had at least one abdominal CT or MRI examination in approximately six months from the US examination. Among them, 10 patients had fatty liver. The diagnostic criteria of fatty liver viewed on CT were: 1) the Hounsfield unit (HU) of the liver was at least 10 HU below that of the spleen, 2) the attenuation of hepatic parenchyma was lower than that of hepatic vasculature, and 3) the attenuation of hepatic parenchyma was < 48 HU [39]. Additionally, the diagnosis of fatty liver viewed on MRI was dependent on a signal drop of at least 10% in out-of-phase imaging than that in in-phase imaging [40].

For patients who experienced cross-sectional imaging within six months of US examination, contrast-enhanced multiphasic CT or liver MRI was used as the reference standard. CT examinations were performed using a multidetector CT scanner with non-contrast, arterial, portal venous, and delayed phases following intravenous contrast administration. MRI examinations included T1-weighted in-phase and opposed-phase imaging, T2-weighted imaging, diffusion-weighted imaging, and dynamic contrast-enhanced sequences when applicable. Diagnostic criteria for LC and focal hepatic lesions were

**Table 2. The scoring system of assessment for image quality.**

| Categories | Score |
|---|---|
| **Brightness** | |
| Difficulty in structure confirmation due to excessive brightness or darkness | 1 |
| Indistinct structure confirmation due to suboptimal brightness in most organs | 2 |
| Indistinct structure confirmation in part of series | 3 |
| Good confirmation in most structure with optimal brightness | 4 |
| Perfect structure confirmation with optimal brightness in all series | 5 |
| **Contrast** | |
| Poor differentiation of between liver and adjacent organs | 1 |
| Poor differentiation of between liver and intrahepatic structure | 2 |
| Indistinct margin of main trunk of portal veins | 3 |
| Good contrast between liver and adjacent organs, but indistinct small intrahepatic vessels | 4 |
| Perfect contrast between liver and adjacent organs or intrahepatic vessels | 5 |
| **Resolution** | |
| Non diagnostic images | 1 |
| Significant blurring of hepatic texture | 2 |
| Acceptable blurring of texture to diagnose space-occupying lesion | 3 |
| Weak blurring texture and distinguishable peripheral vessels within 2 cm of hepatic capsule | 4 |
| Clear image texture | 5 |
| **Reverberation artifact** | |
| Non diagnostic images | 1 |
| Severe | 2 |
| Moderate | 3 |
| Mild | 4 |
| Minimal | 5 |
| **Overall quality** | |
| Non-diagnostic image | 1 |
| Poor | 2 |
| Inadequate images in part of series | 3 |
| Good to diagnosis | 4 |
| Perfect to diagnosis | 5 |

based on established radiologic features, including surface nodularity, segmental atrophy or hypertrophy, portal hypertension findings, and characteristic enhancement patterns.

## Statistical analyses

To assess image quality improvement, we used the paired simple *t* test to assess the five categories mentioned previously. For the inter-reader agreement on the five categories of image quality and diagnostic agreement between two reviewers, we used the weighted kappa value [41]. Kappa value ≤ 0.00 was designated as poor; 0.00–0.20, slight; 0.20–0.40, fair; 0.41–0.60, moderate; 0.61–0.80, substantial; and ≥ 0.81, almost perfect agreement. To compare the diagnostic differences and performance, the McNemar test and receiver operating curve (AUC) analysis were used. We also used MedCalc® software (version 19.0, MedCalc Software, Ostend, Belgium) for all statistical analyses in this study.

**Comparison with other methods.** To validate the proposed method, we compared our method with conventional image filtering method. The author of [16] utilized shock filter to deblur and speckle noise reduction in fetal US image. To

implement it, we used MATLAB tool with mask size 9, and iteration 5. The author of [17] utilized the bilateral filter to liver US image enhancement. They were focused on denoising task. We also implemented it using MATLAB tool.

To further validate the effectiveness of the proposed method, we conducted comparative experiments involving a range of state-of-the-art I2I translation approaches, including contrastive learning–based, transformer-based, and steganography-based models. First, we implemented the Negative Example Generation for Contrastive Learning (NEGCUT) using its original configuration, applying adversarial contrastive learning to layers 1, 5, 9, 13, and 17 of the generator encoder [34]. Second, we reproduced the Negative Sample Pruning method from [35] using the same parameter settings described in the original study. In addition, we implemented UVCGAN [37], which integrates a Unet backbone with Vision Transformer–based attention to enhance long-range structural modeling, and StegoGAN [38], which leverages a steganography-inspired embedding mechanism to support non-bijective translation while preserving structural cues. All comparison models were trained using their recommended training strategies; NEGCUT and Negative Sample Pruning were trained for 25 and 27 epochs, respectively, with a learning rate of 0.0002, while UVCGAN and StegoGAN were trained following the procedures described in their original publications. For each iteration, images were resized and randomly cropped to 384 × 384 to ensure consistent training conditions across methods.

## Results

### Participant characteristics

The mean age with its standard deviation of the included participants for the test dataset (n = 100) was 57 ± 15.17, and the sex ratio was 54: 46 (= male: female). Electronic medical chart review revealed 78 patients with CLD out of 100 test sets (78/100 = 78%), and the etiologies were as follows: chronic hepatitis B virus (HBV) infection (n = 32, 41%), alcoholic liver disease (n = 18, 23%), autoimmune hepatitis (n = 13, 17%), non-alcoholic steatohepatitis (n = 11, 14%), and chronic hepatitis C virus (HCV) infection (n = 4, 5%). While the diagnosis of chronic HBV was based on the persistence of hepatitis B surface antigen for >6 months [42], that of chronic hepatitis C was made when the serum HCV antibody test was positive or HCV RNA was persistently detected 6 months after the onset of acute infection [43]. A total of 47 out of the 78 CLD patients were clinically diagnosed with LC based on laboratory and other imaging tests such as CT and MRI, within 6 months from the liver US. Additionally, the most common cause of LC was chronic HBV infection (n = 21, 45%), followed by alcohol abuse (n = 16, 34%), autoimmune hepatitis (n = 4, 9%), HCV (n = 3, 6%) and non-alcoholic steatohepatitis (n = 3, 6%). Baseline characteristics of the patients of the test dataset are summarized in Table 3.

### Image quality assessment and inter-reader agreement

Brightness, contrast, and overall image quality showed significant improvement as reported by both reviewers in post-processed dataset than those in the original one, with p-values <0.001 (Figs 3–6). However, image resolution and reverberation artifact were not improved for one of both reviewers (p = 0.60 and 0.75, respectively). The assessment of image quality by the two reviewers and the inter-reviewer agreement are summarized in Table 4. The weighted kappa values for all categories of image quality assessment for the two reviewers range from 0.25 to 0.66 in the original, and from 0.28 to 0.49 in post-processed dataset, demonstrating fair to substantial agreement.

### Diagnosis and inter-reader agreement

In the diagnosis of LC, fatty liver, solid hepatic focal lesions, GB polyps, and gallstones, the weighted kappa values ranged from 0.48 to 0.84 in the original, and from 0.43 to 0.68 in post-processed sets for both the reviewers, demonstrating moderate to almost-perfect agreement.

Both reviewers diagnosed significantly more patients with LC at post-processed sets (p = 0.004 and 0.003, respectively) as shown in Fig 7 and Table 5. In terms of diagnostic performance of LC in both reviewers, sensitivity, specificity and 95% confidence interval in original sets were 53.2%, 90.6%, and 0.620–0.804 for reviewer 1; and 59.6%,90.6%, and

**Table 3. Baseline patient characteristics.**

| Characteristics | Number of patients (n = 100) |
|---|---|
| **Age, yr. (mean ± SD)** | 57 ± 15.17 |
| **Sex (Male:Female)** | 54:46 |
| **Smoking** | |
| Never | 68 |
| Ever | 32 |
| **Comorbidities** | |
| Hypertension | 28 |
| Diabetes mellitus | 12 |
| Chronic renal failure | 3 |
| Cardiovascular disease | 6 |
| **Chronic liver disease** | 78 |
| Chronic hepatitis B viral infection | 32 |
| Chronic hepatitis C viral infection | 4 |
| Autoimmune hepatitis | 13 |
| Alcoholic liver disease | 18 |
| Non-alcoholic steatohepatitis | 11 |
| **Liver cirrhosis** | 47 |
| Chronic hepatitis B viral infection | 21 |
| Chronic hepatitis C viral infection | 3 |
| Autoimmune hepatitis | 4 |
| Alcoholic liver disease | 16 |
| Non-alcoholic steatohepatitis | 3 |

*Note: SD = standard deviation.*

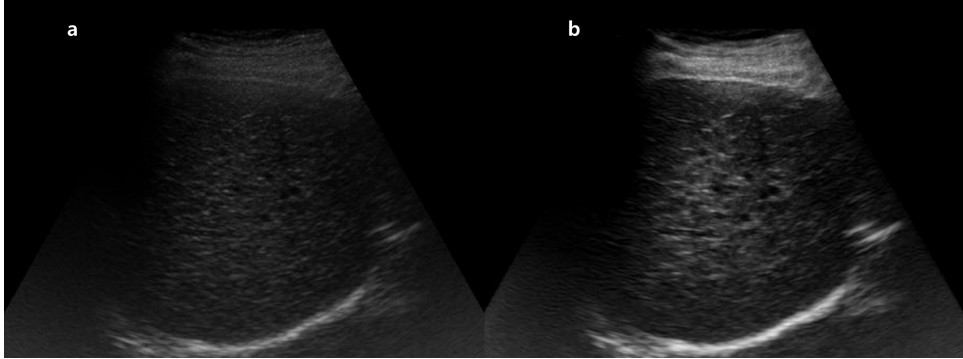

**Fig 3. Representative example of brightness improvement in liver US. (a)** Original right intercostal grey-scale liver US image obtained using a 12-year-old US system. **(b)** Post-processed image generated by the proposed model trained with images from a newer 4-year-old system as the target domain. The enhanced image demonstrates improved global brightness and clearer visualization of hepatic parenchyma. Both reviewers assigned a higher brightness score to the post-processed image (score 4) compared with the original image (score 3). *Note: US, ultrasound.*

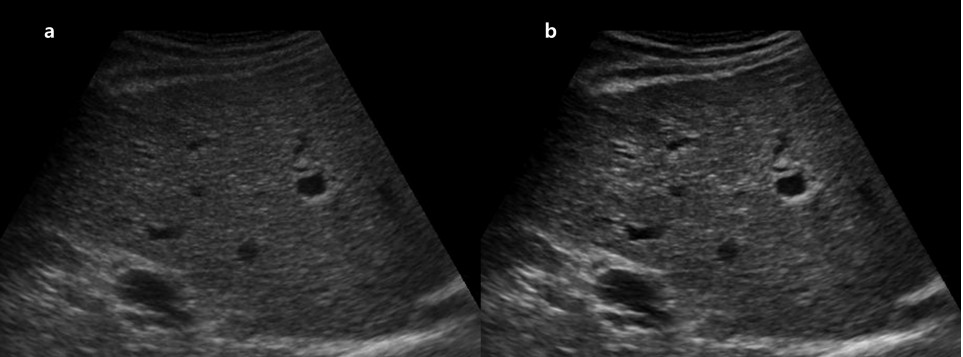

**Fig 4. Representative example of contrast improvement in liver US. (a)** Original right intercostal grey-scale liver US image obtained using a 12-year-old US system. **(b)** Post-processed image generated by the proposed model trained with images from a newer 4-year-old system as the target domain. The enhanced image demonstrates improved contrast between hepatic parenchyma and adjacent structures. Both reviewers assigned higher contrast scores to the post-processed image (score 5) compared with the original image (score 3). *Note: US, ultrasound.*

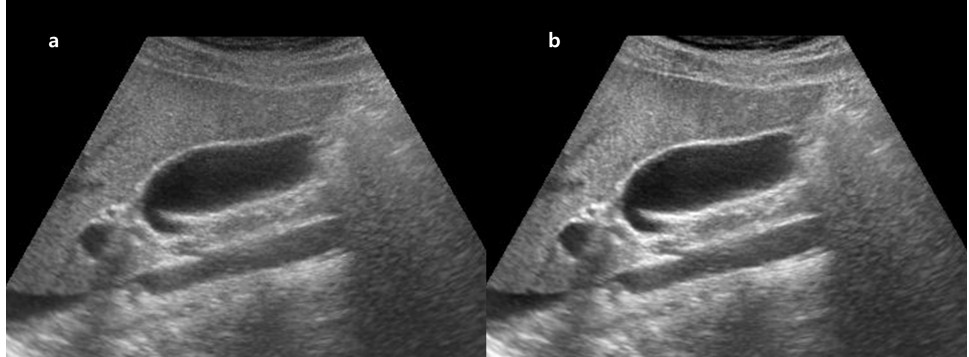

**Fig 5. Representative example of reduced reverberation artifacts in liver US. (a)** Original right intercostal grey-scale liver US image obtained using a 12-year-old US system. **(b)** Post-processed image generated by the proposed model trained with images from a newer 4-year-old system as the target domain. The enhanced image shows reduced near-field reverberation artifacts and improved visualization of hepatic parenchyma. Both reviewers assigned higher reverberation artifact scores to the post-processed image (score 4) compared with the original image (score 3). *Note: US, ultrasound.*

0.654–0.832 for reviewer 2, respectively. Additionally, in post-processed sets, the sensitivity, specificity and 95% confidence interval of LC were 66.0%, 75.5% and 0.608–0.794 for reviewer 1; and 70.2%, 79.2% and 0.651–0.829 for reviewer 2, respectively. No significant difference was identified between the four obtained ROC curves (all p > 0.05).

In 47 patients that had at least one abdominal CT or MRI examination in approximately six months from the US examination, the sensitivity and specificity of fatty liver of the original sets were 50% and 86% for reviewer 1, and 80% and 65% for reviewer 2, respectively. However, in post-processed sets, the sensitivity and specificity of fatty liver were 50% and 89% for reviewer 1, and 80% and 68% for reviewer 2, respectively. The accuracy of reviewers 1 and 2 in the original sets was 0.79 and 0.68, and that in post-processed sets was 0.81 and 0.70 for reviewers 1 and 2, respectively. No significant change between the original and post-processed sets for reviewers was noted (p > 0.05).

Among the 47 patients available for CT or MRI, 10 patients had focal solid hepatic lesions [hepatocellular carcinomas (n = 5), hemangiomas (n = 4), and abscess (n = 1)]. The sensitivity of hepatic focal solid lesions for reviewer 1 was 50% and 60% for original and post-processed sets, respectively, and specificity was 95% and 92%, respectively. Additionally, for reviewer 2, the sensitivity was 70% and 70% in original and post-processed sets, respectively, and specificity was 95%

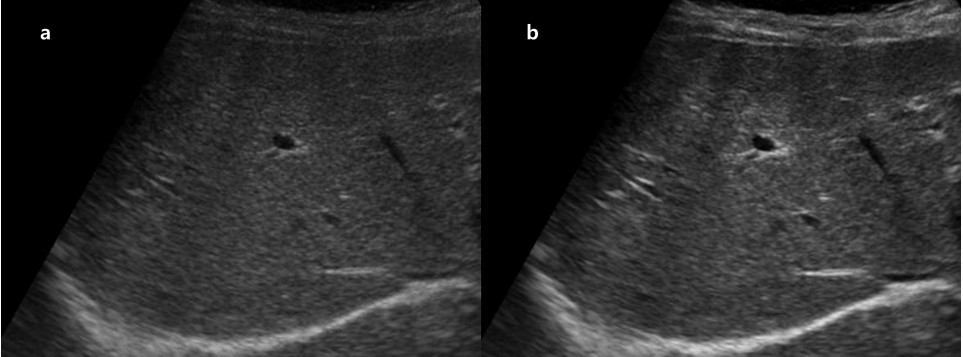

**Fig 6. Representative example of focal lesion detection in liver US.** A 73-year-old woman experienced right intercostal grey-scale liver US examination. **(a)** Original image obtained using a 12-year-old US system. **(b)** Post-processed image generated by the proposed model trained with images from a newer 4-year-old system as the target domain. In the post-processed image, an approximately 2-cm hypoechoic nodule in the right hepatic lobe is more clearly visualized. Both reviewers identified the lesion in the enhanced image, whereas it was not detected in the original image. *Note: US, ultrasound.*

**Table 4. Image quality assessment and inter-reader agreement of reviewer 1 and reviewer 2.**

|  | Original dataset | | | Post-processed dataset | | | *p*-values** | |
|---|---|---|---|---|---|---|---|---|
|  | Reviewer 1 (mean score ±SD) | Reviewer 2 (mean score ±SD) | Inter-reader agreement* | Reviewer 1 (mean score ±SD) | Reviewer 2 (mean score ±SD) | Inter-reader agreement* | Reviewer 1 | Reviewer 2 |
| **Brightness** | 2.87±0.99 | 3.20±0.91 | 0.66 (0.55, 0.76) | 3.38±0.94 | 3.85±0.90 | 0.49 (0.36, 0.63) | <.001 | <.001 |
| **Contrast** | 2.90±0.77 | 3.47±0.91 | 0.39 (0.26, 0.52) | 3.29±0.90 | 4.14±0.79 | 0.28 (0.16, 0.40) | <.001 | <.001 |
| **Resolution** | 2.92±0.73 | 3.67±0.93 | 0.39 (0.24, 0.53) | 3.08±0.76 | 3.71±0.73 | 0.28 (0.12, 0.43) | 0.048 | 0.06 |
| **Reverberation artifact** | 3.19±0.74 | 3.47±0.89 | 0.25 (0.07, 0.42) | 3.22±0.84 | 3.73±0.78 | 0.38 (0.20, 0.55) | 0.75 | <.001 |
| **Overall quality** | 3.00±0.89 | 3.27±0.94 | 0.55 (0.39, 0.71) | 3.36±0.95 | 3.68±0.83 | 0.41 (0.26, 0.57) | <.001 | <.001 |

*Note: SD = standard deviation; * data are weighted kappa values, with 95% confidence intervals in parentheses; **p-values were calculated by paired t test.*

and 89%, respectively. The accuracy of reviewers 1 and 2 in the original sets was 0.85 and 0.89, while that in post-processed sets was 0.85 for both the reviewers. No significant changes between the original and post-processed sets for both reviewers were noted (p > 0.05).

No statistical difference was seen between original and post-processed sets in both reviewer groups (all p > 0.05) for diagnosing GB polyps and stones. Since no imaging modality is superior to US in diagnosing GB polyps and stones, a comparison study of diagnostic performance for GB was not available.

**Algorithm comparison.** The qualitative comparison in Fig 8 demonstrates that the proposed method provides the most consistent and diagnostically meaningful enhancement across all examples. The enhanced outputs show improved brightness and contrast while preserving the underlying hepatic structures. In contrast, the Shock filter and bilateral filter offer minimal visual improvement, showing little change in brightness or parenchymal clarity. NEGCUT and UVCGAN produce noticeable enhancement but excessively smooth fine details that are important for clinical interpretation. Negative Sample Pruning also offers limited improvement, with subtle changes that do not substantially enhance diagnostic

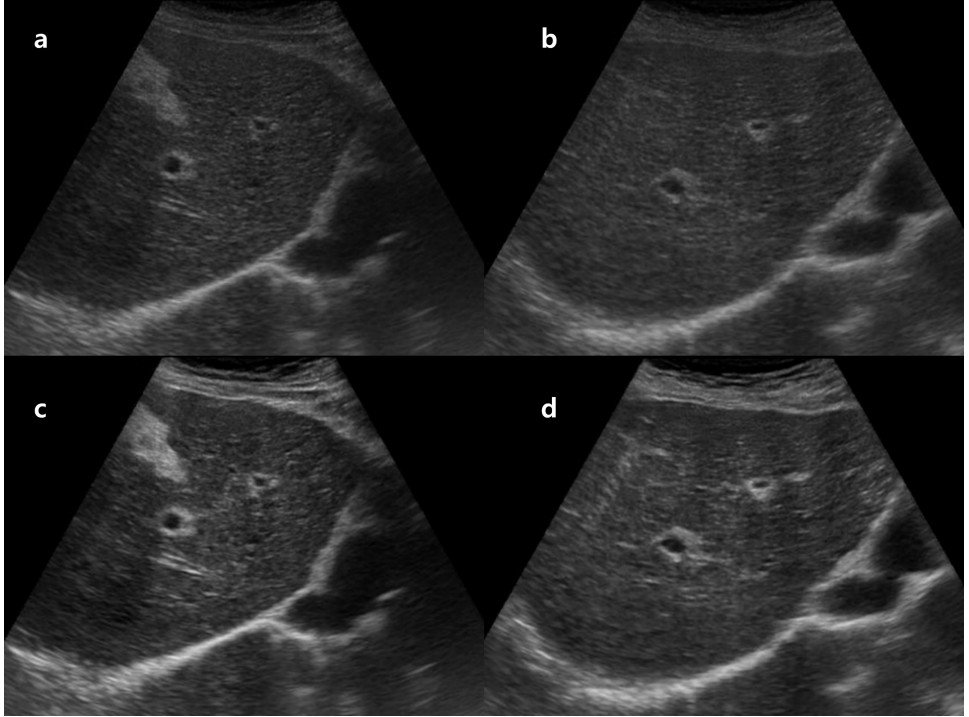

**Fig 7. Representative example of LC detection in US.** A 74-year-old woman with chronic hepatitis C virus infection experienced right subcostal grey-scale liver US examination. **(a, b)** Original images obtained using a 12-year-old US system. **(c, d)** Post-processed images generated by the proposed model trained with images from a newer 4-year-old system as the target domain. In the enhanced images, hepatic surface nodularity is more clearly visualized due to improved near-field contrast. Both reviewers diagnosed LC in the post-processed images, whereas only one reviewer diagnosed LC in the original images. *Note: US, ultrasound; LC, liver cirrhosis.*

**Table 5. Inter-reader agreement of diagnosis between reviewers 1 and 2.**

| | Original dataset | Kappa values* (Original) | Post-processed dataset | Kappa values* (post-processed) | Diagnosis differences between original and post-processed dataset (p-values)** | |
|---|---|---|---|---|---|---|
| | (R1:R2) | (mean score ±SD) | (R1:R2) | (mean score ±SD) | Reviewer 1 | Reviewer 2 |
| **Liver cirrhosis** | 30:33 | 0.84(0.72, 0.95) | 36:44 | 0.68(0.53, 0.82) | 0.0043 | 0.0034 |
| **Fatty liver** | 25:49 | 0.48(0.30, 0.48) | 17:43 | 0.43(0.20, 0.43) | 0.08 | 0.11 |
| **Solid hepatic focal lesions** | 21:17 | 0.61(0.41, 0.81) | 19:15 | 0.51(0.28, 0.73) | 0.79 | 0.63 |
| **Gallbladder polyps** | 12:11 | 0.66(0.42, 0.89) | 9:9 | 0.51(0.22, 0.81) | 0.51 | 0.51 |
| **Gallstones** | 10:12 | 0.51(0.60, 0.99) | 12:13 | 0.59(0.35, 0.93) | 0.73 | 1 |

*Note: R1 = number of patients called from reviewer 1, R2 = number of patients called from reviewer 2, \* data are weighted kappa values, with 95% confidence intervals in parentheses; \*\*p-values were calculated by the McNemar test.*

*SD. Standard deviat.*

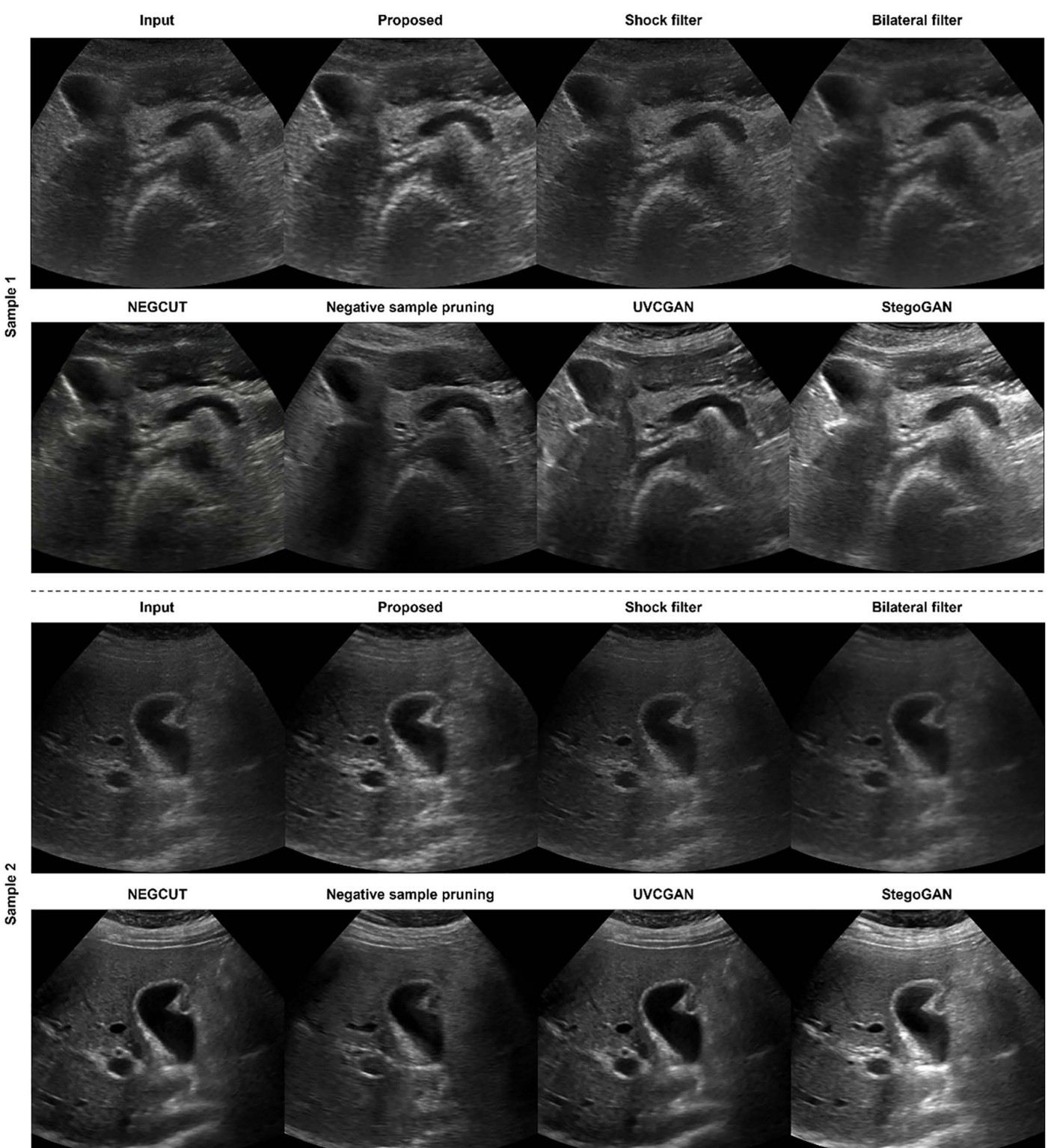

**Fig 8. Qualitative comparison of the proposed method with conventional filtering and state-of-the-art I2I translation approaches.** For each sample, the first row shows results from the input image, the proposed method, Shock filter, and Bilateral filter. The second row shows results from NEG-CUT, Negative Sample Pruning, UVCGAN, and StegoGAN. Two representative liver US samples are presented (Sample 1 and Sample 2). Compared

with conventional filters and other generative models, the proposed method demonstrates balanced enhancement of brightness and contrast while preserving hepatic parenchymal texture and anatomical structures without excessive smoothing or oversaturation.

visibility. StegoGAN generates striking visual enhancement; however, its outputs frequently exhibit oversaturation, resulting in loss of subtle parenchymal textures. Overall, the proposed method achieves a well-balanced enhancement, improving visibility without compromising structural fidelity.

To quantitatively assess the proposed method, we evaluated Contrast Ratio (CR) and Contrast-to-Noise Ratio (CNR) across 100 test samples by measuring foreground and background regions for each image. As shown in Table 6, the proposed method clearly improves CR compared with the input images and achieves a meaningful level of enhancement among the comparison methods. Although the Negative Sample Pruning and StegoGAN method reports the highest CR, this is largely due to excessive saturation in its outputs, which artificially elevates contrast rather than providing diagnostically meaningful enhancement. For CNR, the proposed method demonstrates comparable performance to other approaches and exceeds most methods except the bilateral filter and UVCGAN. The bilateral filter naturally achieves higher CNR due to its inherent noise-suppression mechanism. UVCGAN also reports elevated CNR values; however, this improvement largely results from excessive smoothing, which reduces noise at the expense of fine parenchymal detail. The slight decrease in CNR observed in the proposed method relative to the input reflects the expected trade-off when enhancing contrast while maintaining structural fidelity. Overall, the proposed method achieves a clinically reasonable balance, improving contrast without over smoothing or compromising essential hepatic textures.

To further evaluate the proposed method against other I2I translation approaches, we computed the Fréchet Inception Distance (FID), which measures how closely generated images resemble the target domain. A lower FID score indicates better alignment. Using 1,500 target images and 1,500 generated images for each method, the results are summarized in Table 7. Although StegoGAN achieved the lowest FID score, the proposed method produced a competitive result, demonstrating reasonable generative fidelity relative to the comparison methods. One thing to note here is that the FID is based on an Inception network trained on natural images rather than medical data, therefore its ability to fully capture US characteristics is limited. Despite these constraints, the proposed method provides a balanced performance.

To provide objective evidence that the proposed method preserves diagnostically important structure and texture, we performed a quantitative structure-preservation analysis using patch-wise Structural Similarity Index Measure (SSIM) and multiple first- and second-order statistical metrics in Table 8. Patch-wise SSIM between the input and enhanced images remained high ($0.996 \rightarrow 0.801$), demonstrating that the proposed method preserves structural similarity more effectively than the comparison methods, which showed substantially lower SSIM values. First-order statistics—including mean intensity, standard deviation, skewness, and entropy—showed minimal deviation from the input image, indicating that the intensity distribution and parenchymal roughness were largely maintained. Similarly, second-order Grey-Level Co-occurrence Matrix (GLCM) features such as correlation, energy, and homogeneity—which reflect micro-texture relationships within hepatic parenchyma—remained very close to the input values for the proposed method. The only feature showing a noticeable difference was GLCM contrast. This is expected because GLCM contrast is highly sensitive to local intensity variation and responds strongly when contrast enhancement slightly increases edge sharpness or when subtle speckle components are boosted. Such local changes increase contrast values even when the overall parenchymal texture, as reflected by the other GLCM features, remains preserved.

Taken together, these objective measurements confirm that the proposed method maintains essential hepatic texture characteristics while improving overall image visibility, addressing concerns regarding potential structural alteration or hallucination.

## Discussions

Our study demonstrated the clinical feasibility of unsupervised deep learning to improve the image quality of liver US. Liver US, despite its effectiveness being influenced by the operator's skill and reduced sensitivity in obese or nonviral

**Table 6. Comparison results of CR and CNR value.**

|  | Input | Shock Filter | Bilateral Filter | NEGCUT | Negative Sample Pruning | UVCGAN | StegoGAN | Proposed |
|---|---|---|---|---|---|---|---|---|
| CR | 50.04 | 50.21 | 49.78 | 49.32 | 82.08 | 51.29 | 81.32 | 67.79 |
| CNR | 2.51 | 2.40 | 2.72 | 1.93 | 2.21 | 2.52 | 1.93 | 2.47 |

*Note: Higher CR, CNR value represents better performance.*

**Table 7. Comparison results of FID score.**

|  | NEGCUT | Negative Sample Pruning | UVCGAN | StegoGAN | Proposed |
|---|---|---|---|---|---|
| FID score↓ | 137.50 | 142.53 | 111.60 | 101.94 | 109.81 |

*Note: Lower FID score represents better performance.*

**Table 8. Quantitative evaluation of structure preservation (The parentheses indicate the correlation value between input and each method).**

|  | Input | Shock Filter | Bilateral Filter | NEGCUT | Negative Sample Pruning | UVCGAN | StegoGAN | Proposed |
|---|---|---|---|---|---|---|---|---|
| Patch-wise SSIM | 0.996 | 0.955 (0.99) | 0.953 (0.99) | 0.808 (0.90) | 0.526 (0.58) | 0.479 (0.53) | 0.714 (0.52) | 0.801 (0.84) |
| Mean intensity | 0.259 | 0.265 (0.99) | 0.271 (0.99) | 0.194 (0.76) | 0.154 (0.70) | 0.248 (0.74) | 0.256 (0.52) | 0.250 (0.93) |
| Std | 0.160 | 0.165 (0.99) | 0.166 (0.99) | 0.141 (0.59) | 0.120 (0.54) | 0.153 (0.60) | 0.176 (0.37) | 0.159 (0.85) |
| Skewness | 0.137 | 0.161 (0.99) | 0.088 (0.99) | 0.830 (0.75) | 1.154 (0.30) | 0.371 (0.75) | 0.403 (0.70) | 0.206 (0.99) |
| Entropy | 6.080 | 6.000 (0.99) | 5.982 (0.99) | 6.419 (0.84) | 6.295 (0.83) | 6.397 (0.91) | 6.499 (0.58) | 6.371 (0.99) |
| GLCM Contrast | 0.426 | 0.138 (0.04) | 0.126 (0.33) | 0.397 (0.81) | 0.611 (−0.20) | 0.313 (−0.41) | 0.471 (−0.39) | 0.321 (0.28) |
| GLCM Correlation | 0.958 | 0.985 (0.99) | 0.986 (0.99) | 0.947 (0.98) | 0.918 (0.93) | 0.967 (0.98) | 0.960 (0.97) | 0.968 (0.99) |
| GLCM Energy | 0.748 | 0.732 (0.99) | 0.737 (0.99) | 0.816 (0.76) | 0.824 (0.95) | 0.769 (0.84) | 0.717 (0.52) | 0.744 (0.99) |
| GLCM Homogeneity | 0.988 | 0.994 (0.99) | 0.994 (0.99) | 0.989 (0.99) | 0.985 (0.99) | 0.990 (0.99) | 0.988 (0.99) | 0.990 (0.99) |

liver disease patients, remains essential for LC and subsequent HCC screening. Although quantitative statistical data is lacking globally, it is widely recognized that in developing countries, there is a prevalence of outdated US equipment. On the other hand, improving image quality in essential ultrasonography has been a very important challenge and has continued to advance to the present day. Since the introduction of deep learning technology to US, several studies have applied this technology to liver US to improve the diagnostic performance of focal liver lesions or diffuse liver diseases, such as fatty liver or fibrosis [22–25,44–46]. However, as far as we investigated, no attempt to improve the image quality of liver US via deep learning is known. Furthermore, our study is significant in that it attempts a quantitative evaluation of essential aspects of liver US such as brightness, contrast, and resolution in the realm of image quality improvement. Our results showed significant image quality improvement in brightness, contrast, and overall quality for both reviewers.

However, one of the two reviewers unexpectedly rated similar scores for resolution and reverberation artifact categories. Since US is an operator-dependent study and has various individual preferences in terms of image texture and setting parameters, the criteria for image quality are more subjective than those for other imaging modalities such as CT or MRI. Therefore, slightly low kappa values from inter-reader agreement tests of image quality assessment, i.e., 0.25 to 0.66, were understandable. Additionally, we had to be careful and minimize adjustment of the image texture because of possible misdiagnosis of CLD and LC. Therefore, the degree of improvement in image resolution with our algorithm may seem to be minimal.

Although no difference in overall AUCs for LC was noted for both reviewers, > 10% increase in sensitivity of post-processed sets for both reviewers is another remarkable finding in this study. This gain in sensitivity was accompanied by an approximately 15–16% decrease in specificity, and there was no significant improvement in overall diagnostic accuracy as assessed by ROC analysis. This sensitivity–specificity trade-off implies that our post-processed images may increase the number of false-positive findings for LC, which could potentially lead to additional follow-up examinations or unnecessary patient anxiety. Therefore, the observed improvement in sensitivity should be interpreted with caution, as it does not necessarily translate into better overall diagnostic performance or improved patient outcomes. In this study, our primary goal was to explore the technical feasibility of unsupervised deep learning–based image quality enhancement and its potential impact on LC detection, and a more optimized operating point that balances sensitivity and specificity will need to be investigated in future work. As indicated by the results, particularly noteworthy is the superior mean score in contrast and resolution within the post-processed set. Furthermore, a trend toward the reduction of reverberation artifacts in the near zone was observed in the post-processed imaging, caused by abdominal wall layers and peritoneum. Consequently, it can be inferred that surface nodularity, an important imaging indicator of LC, was more effectively detected by both reviewers.

Between sensitivity and specificity, higher sensitivity is more important than specificity because early detection of LC for early intervention is the major purpose of imaging surveillance in CLD patients, along with screening for hepatocellular carcinomas. However, the sensitivity of US for LC without elastography (52–69%) is significantly lower than that of CT or MRI (77–84%), which is a major drawback of liver US [20,26]. Thus, complementing the low sensitivity of US for LC, our proposed method could be a solution to maintain US as a first-line modality for diagnosing liver disease.

No difference was seen between the reviewers' observations as well as between original and post-processed sets regarding the sensitivities and specificities of fatty liver and solid hepatic focal lesions. Although our study involved only 10 patients with solid hepatic focal lesions out of 47 patients with available abdominal CT or MRI, our deep learning approach hardly deteriorated internal structures and their own echogenicity, a major concern of deep learning-based image reconstruction.

We also observed several situations in which the proposed enhancement was suboptimal. In cases with very severe near-field reverberation artifacts caused by the abdominal wall or marked rib shadowing, the algorithm sometimes failed to sufficiently suppress artifacts or even slightly accentuated them, leading to noisy appearances along the liver surface, as shown in Fig 9. In some obese patients or those with markedly heterogeneous parenchyma, the enhancement of contrast and brightness did not consistently translate into clearer depiction of surface nodularity or vascular structures. These failure cases suggest that the model is less robust in the presence of extreme artifacts and challenging patient anatomy, and that additional artifact-aware training strategies or tailored post-processing may be required before broader clinical deployment.

In this study, we adopted a Switchable CycleGAN framework to translate a low- to high-quality liver US image in an unsupervised manner to support the diagnostic accuracy for LC. Although GAN-based models have demonstrated strong performance in medical image enhancement, concerns remain regarding the potential generation of artificial patterns or hallucinated pathology. We specifically selected the Switchable CycleGAN architecture to mitigate this risk. Unlike the conventional CycleGAN framework that employs two independent generators for bidirectional translation, the Switchable

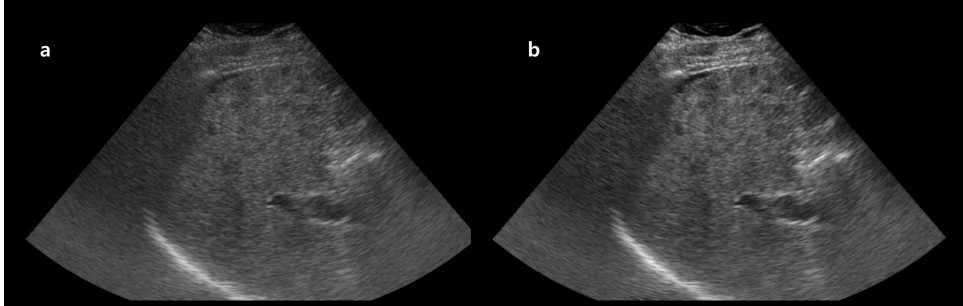

**Fig 9. Representative images for failure case due to heterogeneous parenchyma. (a)** Original right intercostal grey-scale liver US image obtained using a 12-year-old US system. **(b)** Post-processed image generated by the proposed model trained with images from a newer 4-year-old system as the target domain.

CycleGAN utilizes a single shared generator whose transformation direction is controlled by AdaIN-based domain codes. This shared-parameter design constrains the forward and backward mappings within a unified feature representation, thereby reducing the degrees of freedom that may otherwise allow unintended structural alterations. Furthermore, the AdaIN mechanism modulates channel-wise feature statistics (mean and variance) rather than introducing new spatial structures, which makes the enhancement process closer to contrast and texture normalization rather than anatomical synthesis. In addition, cycle-consistency and identity losses were imposed during training to discourage structural deformation and unnecessary modification of clinically meaningful regions.

To empirically verify that pathology was not altered, we conducted quantitative structure-preservation analysis using patch-wise SSIM and first- and second-order texture statistics. The proposed method maintained high structural similarity to the input images and preserved hepatic micro-texture features, as shown in Table 8. Importantly, no statistically significant changes were observed in the diagnostic performance of focal hepatic lesions or fatty liver detection, suggesting that the enhancement process did not artificially create or obscure pathology.

Taken together, both architectural design and quantitative validation support that the proposed method performs controlled contrast enhancement while preserving clinically relevant anatomical structures.

The observed inference time suggests the feasibility of near real-time deployment during US scanning. The feed-forward architecture of the CycleGAN-based model allows image enhancement without iterative reconstruction, making it suitable for integration into clinical workflows. With GPU acceleration, processing speeds of approximately 25–30 FPS are achievable, which is comparable to standard US frame acquisition rates. Further optimization, model compression, and integration into dedicated hardware could enable real-time implementation directly on US systems.

Our study has some limitations. First, its retrospective design was unavoidable because substantial time and resources were required to develop and train the deep learning algorithm. Moreover, the enhancement method improves visualization, the current study does not evaluate real-time deployment. Prospective studies and further optimization of inference speed will be necessary to enable point-of-care implementation. Second, the criterion for categorizing low- and high-quality images in this study was only the age of the US machines. It is possible that an old US machine maintains good image quality under proper management. However, in general, older machines show a lower quality of images over time. Furthermore, for an individual, classifying the quality of thousands of US studies would be subjective and time-consuming; therefore, we used the age of the machines as the criterion for the quality of US. Third, the model was trained using only one pair of US systems, effectively performing device-to-device translation rather than universally generalizable quality enhancement. Because US image appearance is strongly influenced by vendor-specific beamforming algorithms, dynamic range compression, and proprietary post-processing pipelines, image statistics may differ substantially across

manufacturers. Therefore, direct application of the current model to images acquired from other vendors may not guarantee consistent performance.

In practice, adaptation to new devices would likely require retraining or fine-tuning using unpaired dataset from the corresponding vendor. Importantly, because our framework operates in an unsupervised manner and does not require paired ground-truth images, domain adaptation can be achieved with relatively modest data requirements. Furthermore, future work may extend the current approach to a multi-domain training setting, where images from multiple vendors are jointly learned within a shared latent representation. Such a strategy could improve generalizability and move the method from device-specific translation toward vendor-agnostic quality normalization.

Fourth, although the test dataset was randomly shuffled, radiologists may not have been fully blinded to whether an image was original or post-processed. GAN-enhanced images may exhibit subtle visual characteristics that experienced readers could potentially recognize, introducing unintentional bias into subjective quality scoring. This limitation has been acknowledged, and future prospective studies will incorporate explicit blinding procedures to minimize observer bias.

Fifth, In the present study, the relatively small sample size of the non-CLD group (n = 22) within the validation set (n = 100) should be acknowledged as an important limitation, as it may reduce the statistical power to detect potential false cirrhosis-like texture generation in livers without chronic liver disease.

Finally, cross-sectional imaging (CT or MRI) was accepted within a six-month interval from the US examination. Although this timeframe reflects common clinical surveillance intervals for CLD, it may introduce potential discrepancies due to interval disease progression. However, the evaluated conditions—such as LC and fatty liver—are generally chronic structural abnormalities that evolve gradually rather than acutely. Nevertheless, the possibility of interval change cannot be entirely excluded, and future prospective studies with shorter temporal intervals would provide stronger validation.

## Conclusion

The proposed algorithm retrospectively enhances the low-quality liver US images to high quality, thereby enabling clearer visualization that may assist clinical interpretation and intervention for LC. Future work will focus on validating the model in multicenter settings with diverse US vendors and protocols, as well as exploring integration with other quantitative tools (e.g., elastography or radiomics) to further improve the detection of CLD and related complications.

## Author contributions

**Conceptualization:** Eun Sun Lee, Jong Chul Ye.

**Data curation:** Jaeyoung Huh, Joo Hyeok Choi.

**Formal analysis:** Joo Hyeok Choi.

**Investigation:** Jaeyoung Huh, Joo Hyeok Choi, Eun Sun Lee, Jeong Eun Lee, Hyun Jeong Park, Byung Ihn Choi.

**Methodology:** Jaeyoung Huh, Joo Hyeok Choi, Eun Sun Lee, Jong Chul Ye.

**Supervision:** Eun Sun Lee, Jong Chul Ye.

**Writing – original draft:** Jaeyoung Huh, Joo Hyeok Choi.

**Writing – review & editing:** Eun Sun Lee, Jong Chul Ye, Jeong Eun Lee, Hyun Jeong Park, Byung Ihn Choi.

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
