## [Decision Letter · Decision Letter 0]

22 Jul 2024

PONE-D-23-32222Image quality improvement of liver ultrasound using unsupervised deep learningPLOS ONE

Dear Dr. Lee,

Thank you for submitting your manuscript to PLOS ONE. After careful consideration, we feel that it has merit but does not fully meet PLOS ONE’s publication criteria as it currently stands. Therefore, we invite you to submit a revised version of the manuscript that addresses the points raised during the review process.

We look forward to receiving your revised manuscript.

Kind regards,

Muhammad Ramzan, PhD

Academic Editor

PLOS ONE

Journal Requirements:

Reviewers' comments:

Reviewer's Responses to Questions

**Comments to the Author**

1. Is the manuscript technically sound, and do the data support the conclusions?

Reviewer #1: Partly

Reviewer #2: Partly

2. Has the statistical analysis been performed appropriately and rigorously? 

Reviewer #1: I Don't Know

Reviewer #2: No

3. Have the authors made all data underlying the findings in their manuscript fully available?

The PLOS Data policy requires authors to make all data underlying the findings described in their manuscript fully available without restriction, with rare exception (please refer to the Data Availability Statement in the manuscript PDF file). The data should be provided as part of the manuscript or its supporting information, or deposited to a public repository. For example, in addition to summary statistics, the data points behind means, medians and variance measures should be available. If there are restrictions on publicly sharing data—e.g. participant privacy or use of data from a third party—those must be specified.requires authors to make all data underlying the findings described in their manuscript fully available without restriction, with rare exception (please refer to the Data Availability Statement in the manuscript PDF file). The data should be provided as part of the manuscript or its supporting information, or deposited to a public repository. For example, in addition to summary statistics, the data points behind means, medians and variance measures should be available. If there are restrictions on publicly sharing data—e.g. participant privacy or use of data from a third party—those must be specified.requires authors to make all data underlying the findings described in their manuscript fully available without restriction, with rare exception (please refer to the Data Availability Statement in the manuscript PDF file). The data should be provided as part of the manuscript or its supporting information, or deposited to a public repository. For example, in addition to summary statistics, the data points behind means, medians and variance measures should be available. If there are restrictions on publicly sharing data—e.g. participant privacy or use of data from a third party—those must be specified.requires authors to make all data underlying the findings described in their manuscript fully available without restriction, with rare exception (please refer to the Data Availability Statement in the manuscript PDF file). The data should be provided as part of the manuscript or its supporting information, or deposited to a public repository. For example, in addition to summary statistics, the data points behind means, medians and variance measures should be available. If there are restrictions on publicly sharing data—e.g. participant privacy or use of data from a third party—those must be specified.

Reviewer #1: No

Reviewer #2: No

4. Is the manuscript presented in an intelligible fashion and written in standard English?

Reviewer #1: Yes

Reviewer #2: Yes

5. Review Comments to the Author

Reviewer #1: 1. Improve the quality of Fig 2. The text is not in readable form.

2. Verify that the format of the abstract is according to the journal format.

3. References format is not according to the journal format.

4. Improved the Flow chart of liver ultrasound dataset preparation.

5. Some experiments will be done which show the results before and after Image quality improvement of liver ultrasound.

6. How does the proposed unsupervised deep learning approach compare with existing methods for liver ultrasound image enhancement

7. Need to enhanced the Introduction section.

8. Can you provide a more detailed explanation of the network architecture used in the study?

Reviewer #2: The manuscript does not adhere to our journal's formatting guidelines. It has been found that the figures and tables placement within the manuscript is incorrect, and the captions are improperly formatted. References are not according properly. Additionally, the quality and resolution of the figures do not meet our publication standards. Need to improve the abstract, introduction section, and methodology. Need to add some more discussion about the background and existing work. Add the dataset description in tabular form. Some Mathematical equations have been used but do not mention the equation number. Add some more details about how to utilize the Switchable cycle generative adversarial network. As the proposed method was evaluated by visual inspection, is it better some other way will be used for the analysis and evaluated the images?

6. PLOS authors have the option to publish the peer review history of their article (what does this mean?). If published, this will include your full peer review and any attached files.). If published, this will include your full peer review and any attached files.). If published, this will include your full peer review and any attached files.). If published, this will include your full peer review and any attached files.

...

Reviewer #1: No

Reviewer #2: No

---

## [Author Response · Author response to Decision Letter 1]

14 Sep 2024

August 28th, 2024

Editorial office, PLOS ONE

We would like to thank you and the reviewers for the valuable and constructive comments offered on our previous submission entitled " Image quality improvement of liver ultrasound using unsupervised deep learning" to the PLOS ONE. We prepared a point-by-point response to the reviewers’ comments. We tried to accept all the editor and reviewers' suggestions and answer all queries. Our specific responses to indications and suggestions are as follows:

To the comments of Reviewer 1

1. Improve the quality of Fig 2. The text is not in readable form.

Thank you for your comments. As reviewer 1’s comment, we have improved the image quality of Figure 2 as you suggested, and I have adjusted the font size. Thank you for your thoughtful comment.

2. Verify that the format of the abstract is according to the journal format.

Thank you for your comments. We revised the format of the abstract adheres to the journal’s guide lines. Thank you for your attention to detail and grateful for the opportunity to ensure that our submission meets the required standards

3. References format is not according to the journal format.

Thank you for your descent comments. Following the reviewer’s advice, we have revised the references.

4. Improved the Flow chart of liver ultrasound dataset preparation.

Thank you for your comments. We revised our flow chart of the liver ultrasound dataset preparation to enhance its visibility and clarity as follow.:

5. Some experiments will be done which show the results before and after Image quality improvement of liver ultrasound.

Thank you for your comments. From our understanding, we have to do some comparison experiments and add the results of it with the proposed method results. Therefore, we have added Figure 8, which includes our additional results along with those from other comparison experiments. If we have misunderstood your focus, please let us know, and we will make further revisions accordingly.

6. How does the proposed unsupervised deep learning approach compare with existing methods for liver ultrasound image enhancement

Thank you for your comments. Conventional liver ultrasound image enhancement methods rely on various image filters. To verify that our proposed method is superior, we implemented the shark and bilateral filters, as used in references (43, 44), and included the results in Figure 8. Our method is based on the image-to-image (I2I) translation approach, which has been widely researched. We conducted a comparative experiment using a contrastive-based learning method for efficient I2I, training the open-source code provided by the authors on the same dataset as our method. The results are discussed in the section 'Comparison with Other Methods', ’Algorithm comparison’ section. For a quantitative evaluation, we calculated the CR and CNR values to assess contrast enhancement and the Frechet Inception Distance (FID) score to measure the distance between the generated images and real target images in the feature domain. Our results show that the proposed method performs similarly to or better than other methods both qualitatively and quantitatively.

7. Need to enhanced the Introduction section.

Thank you for your comments. We have refined the introduction section through the following sentences to enhance its comprehensibility.

8. Can you provide a more detailed explanation of the network architecture used in the study?

Thank you for your comments. In ‘Swithcable cycle generative adversarial network’ section, we revised and added more details about the entire framework of the Switchable CycleGAN and the generator architecture, including Adaptve Instance Normalization (AdaIN) Code Generator (ACG).

To the comments of Reviewer 2

The manuscript does not adhere to our journal's formatting guidelines. It has been found that the figures and tables placement within the manuscript is incorrect, and the captions are improperly formatted. References are not according properly. Additionally, the quality and resolution of the figures do not meet our publication standards. Need to improve the abstract, introduction section, and methodology. Need to add some more discussion about the background and existing work. Add the dataset description in tabular form. Some Mathematical equations have been used but do not mention the equation number. Add some more details about how to utilize the Switchable cycle generative adversarial network. As the proposed method was evaluated by visual inspection, is it better some other way will be used for the analysis and evaluated the images?

The abstract, introduction and method sections was revised to enhanced clarity and comprehensibility. We have expanded the discussion on existing work and background to provide more thorough context. A table summarizing the dataset preparation was added to the manuscript.

Evaluation methods for ultrasound image quality can vary widely. However, the assessment of image quality itself may not always align with clinical significance. Therefore, our study posits that evaluations performed through visual assessment by experienced radiologists, especially abdominal radiologists directly involved in clinical practice hold greater relevance.

1. The manuscript does not adhere to our journal's formatting guidelines. It has been found that the figures and tables placement within the manuscript is incorrect, and the captions are improperly formatted. References are not according properly. Additionally, the quality and resolution of the figures do not meet our publication standards.

Thank you for your descent comments. We revised the manuscript to ensure compliance with the guidelines. The placement of graphs and tables has been corrected, and the caption formatting adjusted accordingly. Also, we properly formatted the references to align with the requirements. The quality and resolution of the figures was improved to meet the publication standards.

2. Need to improve the abstract, introduction section, and methodology.

Thank you for your descent comments. The abstract, introduction and method sections was revised to enhanced clarity and comprehensibility.

3. Need to add some more discussion about the background and existing work.

Thank you for the constructive comments. We added contents about existing work about medical image quality enhancement and image translation method in deep learning technique on ‘Introduction’ section.

4. Add the dataset description in tabular form.

Thank you for your comments. We added a table for dataset in the Material and Methods section to aid in its comprehension. Should there be any point for improvement, we will make the necessary revision. Thank you again your comment.

5. Some Mathematical equations have been used but do not mention the equation number.

Thank you for comments. We added the equation number.

6. Add some more details about how to utilize the Switchable cycle generative adversarial network.

Thank you for valuable comment. In ‘Switchable cycle generative adversarial network’ section, we added and revised more information about the Switchable CycleGAN framework.

7. As the proposed method was evaluated by visual inspection, is it better some other way will be used for the analysis and evaluated the images?

Thank you for the constructive comments. One of our key metrics for image quality assessment is contrast. Therefore, we included the Contrast Ratio (CR) and Contrast Noise Ratio (CNR) values for a quantitative comparison between the proposed method and other methods. We selected foreground and background regions in the images and calculated these values, averaging the results over 100 test sets, as shown in Table 6. Since our proposed method is trained in an unsupervised manner and does not use an exact paired dataset, we were unable to perform a direct quantitative evaluation. However, in image style transfer research, the Frechet Inception Distance (FID) score is commonly used to evaluate how closely the network's output resembles real target images. The FID score measures the distribution distance between real and generated images. For this study, we utilized 1,500 real high-quality images and 1,500 generated high-quality images to calculate the FID score, with the results presented in Table 5. Note that the FID score was utilized only for I2I translation comparison methods.

Thank you again for the valuable comments of the reviewers. We did our best for the revision to enhance our manuscript's quality. We hope that the revised version of the manuscript meets with your and reviewer’s approval and will be considered for publication in the PLOS ONE

Sincerely yours,

Eun Sun Lee, MD, PhD

---

## [Decision Letter · Decision Letter 1]

23 Oct 2025

PONE-D-23-32222R1Image quality improvement of liver ultrasound using unsupervised deep learningPLOS ONE

Dear Dr. Lee,

Thank you for submitting your manuscript to PLOS ONE. After careful consideration, we feel that it has merit but does not fully meet PLOS ONE’s publication criteria as it currently stands. Therefore, we invite you to submit a revised version of the manuscript that addresses the points raised during the review process.

Specifically, there were severe issues raised by the reviewers regarding  detail from introduction to discussion.

We look forward to receiving your revised manuscript.

Kind regards,

Do Young Kim, MD, PhD

Academic Editor

PLOS ONE

Journal Requirements:

Reviewers' comments:

Reviewer's Responses to Questions

**Comments to the Author**

1. If the authors have adequately addressed your comments raised in a previous round of review and you feel that this manuscript is now acceptable for publication, you may indicate that here to bypass the “Comments to the Author” section, enter your conflict of interest statement in the “Confidential to Editor” section, and submit your "Accept" recommendation.

Reviewer #3: (No Response)

Reviewer #4: (No Response)

2. Is the manuscript technically sound, and do the data support the conclusions?

Reviewer #3: No

Reviewer #4: Partly

3. Has the statistical analysis been performed appropriately and rigorously? 

Reviewer #3: Yes

Reviewer #4: Yes

4. Have the authors made all data underlying the findings in their manuscript fully available?

The PLOS Data policy requires authors to make all data underlying the findings described in their manuscript fully available without restriction, with rare exception (please refer to the Data Availability Statement in the manuscript PDF file). The data should be provided as part of the manuscript or its supporting information, or deposited to a public repository. For example, in addition to summary statistics, the data points behind means, medians and variance measures should be available. If there are restrictions on publicly sharing data—e.g. participant privacy or use of data from a third party—those must be specified.requires authors to make all data underlying the findings described in their manuscript fully available without restriction, with rare exception (please refer to the Data Availability Statement in the manuscript PDF file). The data should be provided as part of the manuscript or its supporting information, or deposited to a public repository. For example, in addition to summary statistics, the data points behind means, medians and variance measures should be available. If there are restrictions on publicly sharing data—e.g. participant privacy or use of data from a third party—those must be specified.requires authors to make all data underlying the findings described in their manuscript fully available without restriction, with rare exception (please refer to the Data Availability Statement in the manuscript PDF file). The data should be provided as part of the manuscript or its supporting information, or deposited to a public repository. For example, in addition to summary statistics, the data points behind means, medians and variance measures should be available. If there are restrictions on publicly sharing data—e.g. participant privacy or use of data from a third party—those must be specified.requires authors to make all data underlying the findings described in their manuscript fully available without restriction, with rare exception (please refer to the Data Availability Statement in the manuscript PDF file). The data should be provided as part of the manuscript or its supporting information, or deposited to a public repository. For example, in addition to summary statistics, the data points behind means, medians and variance measures should be available. If there are restrictions on publicly sharing data—e.g. participant privacy or use of data from a third party—those must be specified.

Reviewer #3: No

Reviewer #4: Yes

5. Is the manuscript presented in an intelligible fashion and written in standard English?

Reviewer #3: No

Reviewer #4: Yes

6. Review Comments to the Author

Reviewer #3: Image Quality Improvement of Liver Ultrasound Using Unsupervised Deep Learning

This study utilizes a deep learning approach with a neural network based on a switchable cycle generative adversarial network (GAN) to enhance liver ultrasound image quality. The model was trained in an unsupervised setting, using low-quality images from an older device as inputs and high-quality images from a newer device as targets. However, this study still has some major issues such as:

1. Heading after heading is not a good approach add content after each heading that covers what are you discussing in section: Like heading

Background (add content) then start next subheadings

2. Quality of the images, particularly Fig 2, is inadequate. Both reviewers have noted this issue, and improvements are necessary to ensure that all figures are clearly readable

3. The motivation behind this study is not clearly articulated. It is essential to express why this research is significant and what specific problems it aims to address.

4. To strengthen the introduction, please include a bullet-point list of the study's contributions before concluding this section. This will help readers understand the unique aspects of your work.

5. Clearly articulate the research gap that this study intends to fill. This will enhance the clarity and purpose of the study.

6. After recommendations of both reviewers, I am still on to improve introduction section. The introduction is not strong and can be improved.

7. A dedicated section on related work should be added to explore various methodologies for improving detection quality from liver ultrasound images, explore from traditional methods machine learning methods to advanced deep learning including transfer learning.

8. The related work section lacks references to recent studies (2023 and 2024). It is crucial to include and discuss these works to demonstrate the current landscape of research in this area.

9. Please include a comparison of your proposed results with recent existing studies. This will provide context for the significance of your findings.

10. There should be a separate section for conclusions and future directions. In this section, clearly state the limitations of your work to provide a balanced view of your research.

Reviewer #4: Manuscript #PONE-D-23-32222R1

Manuscript title: Image quality improvement of liver ultrasound using unsupervised deep learning

Summary:

This manuscript by Huh et al. presents a deep learning approach using a Switchable CycleGAN model to enhance low-quality liver ultrasound images obtained from an older scanner, using high-quality images from a newer device as the target domain. The authors report that the algorithm improves brightness, contrast, and overall image quality, as evaluated by two radiologists, and modestly increases the sensitivity for detecting liver cirrhosis. The study addresses an important technical challenge of image quality heterogeneity among ultrasound devices and introduces an unsupervised learning framework that eliminates the need for paired datasets. The manuscript is generally well organized and technically sound. However, several aspects regarding the intended clinical application and translational significance of this work remain unclear and should be clarified to avoid potential overinterpretation.

Comments

1. The manuscript does not clearly specify whether the proposed algorithm is intended for retrospective post-hoc enhancement of previously acquired ultrasound images or for real-time enhancement during ongoing image acquisition. Based on the study design, dataset composition, and evaluation procedure, the present work clearly represents a retrospective post-processing approach. All data used for training and testing were previously collected static ultrasound images obtained between 2016 and 2018 from an older system, and the performance evaluation was conducted by comparing original and AI-processed images offline. Therefore, the algorithm has not been validated or demonstrated in a real-time or prospective scanning setting.

However, several sentences in the abstract and discussion—particularly the claim that the algorithm "facilitates early diagnosis and intervention for liver cirrhosis” could be interpreted as implying real-time diagnostic benefit. This creates a conceptual inconsistency between the actual experimental setup and the implied clinical utility. The authors should explicitly clarify that the current study was conducted retrospectively on stored static images and that real-time clinical application remains a future goal requiring further technical development, such as model optimization for inference speed, hardware integration, and prospective validation. Clear distinction between retrospective enhancement and potential real-time deployment is essential to properly contextualize the clinical relevance of this work.

2. The manuscript highlights a >10% increase in sensitivity for liver cirrhosis but fails to address the concurrent 15-16% decrease in specificity. With no significant improvement in overall diagnostic accuracy (ROC curves), this trade-off could lead to more false positives, undermining the claim of clinical benefit. The authors should expand the Discussion to critically address this sensitivity-specificity trade-off, acknowledging that the reported sensitivity gain does not necessarily translate to improved diagnostic accuracy or patient outcomes.

3. The manuscript lacks a discussion of the algorithm's limitations by only presenting successful outcomes. To provide a more balanced and clinically relevant assessment, the authors should include a failure analysis, discussing and showing examples where the image enhancement was suboptimal or failed, particularly in cases with severe artifacts or challenging patient anatomy.

4. The evaluation of image quality relies heavily on subjective radiologist scores. Objective metrics (CR, CNR) were used only for algorithmic comparison, not for validating the clinical test images. Crucially, the study lacks objective proof that the GAN preserves diagnostic vital parenchymal textures, which is a key concern for AI-generated medical images.

5. The model's applicability is highly limited as it was trained on only one specific pair of ultrasound machines (Aloka to Canon). It functions more as a device-to-device translator than a universally applicable enhancement tool. Its performance on equipment from other vendors is unproven and likely poor.

6. While the authors state that test datasets were "randomly shuffled," it is unclear whether the radiologists were blinded to image type (original vs. post-processed). Visual characteristics introduced by the GAN might allow experienced readers to identify processed images, potentially biasing subjective quality ratings. The authors should clarify the blinding procedure or, if blinding was not ensured, acknowledge this as a limitation.

7. PLOS authors have the option to publish the peer review history of their article (what does this mean?). If published, this will include your full peer review and any attached files.). If published, this will include your full peer review and any attached files.). If published, this will include your full peer review and any attached files.). If published, this will include your full peer review and any attached files.

...

Reviewer #3: No

Reviewer #4: No

---

## [Author Response · Author response to Decision Letter 2]

10 Dec 2025

Thanks for both reviewer's important and kind comment. We aimed to address the reviewer’s comments as thoroughly as possible and revised the manuscript accordingly.

---

## [Decision Letter · Decision Letter 2]

25 Jan 2026

PONE-D-23-32222R2Image quality improvement of liver ultrasound using unsupervised deep learningPLOS One

Dear Dr. Lee,

Thank you for submitting your manuscript to PLOS ONE. After careful consideration, we feel that it has merit but does not fully meet PLOS ONE’s publication criteria as it currently stands. Therefore, we invite you to submit a revised version of the manuscript that addresses the points raised during the review process.

Although the revised manuscript is well-written, methodological clarification is still required.

We look forward to receiving your revised manuscript.

Kind regards,

Do Young Kim, MD, PhD

Academic Editor

PLOS One

Journal Requirements:

Reviewers' comments:

Reviewer's Responses to Questions

**Comments to the Author**

1. If the authors have adequately addressed your comments raised in a previous round of review and you feel that this manuscript is now acceptable for publication, you may indicate that here to bypass the “Comments to the Author” section, enter your conflict of interest statement in the “Confidential to Editor” section, and submit your "Accept" recommendation.

Reviewer #4: (No Response)

Reviewer #5: (No Response)

2. Is the manuscript technically sound, and do the data support the conclusions?

Reviewer #4: Yes

Reviewer #5: Partly

3. Has the statistical analysis been performed appropriately and rigorously? 

Reviewer #4: Yes

Reviewer #5: No

4. Have the authors made all data underlying the findings in their manuscript fully available?

The PLOS Data policy requires authors to make all data underlying the findings described in their manuscript fully available without restriction, with rare exception (please refer to the Data Availability Statement in the manuscript PDF file). The data should be provided as part of the manuscript or its supporting information, or deposited to a public repository. For example, in addition to summary statistics, the data points behind means, medians and variance measures should be available. If there are restrictions on publicly sharing data—e.g. participant privacy or use of data from a third party—those must be specified.requires authors to make all data underlying the findings described in their manuscript fully available without restriction, with rare exception (please refer to the Data Availability Statement in the manuscript PDF file). The data should be provided as part of the manuscript or its supporting information, or deposited to a public repository. For example, in addition to summary statistics, the data points behind means, medians and variance measures should be available. If there are restrictions on publicly sharing data—e.g. participant privacy or use of data from a third party—those must be specified.requires authors to make all data underlying the findings described in their manuscript fully available without restriction, with rare exception (please refer to the Data Availability Statement in the manuscript PDF file). The data should be provided as part of the manuscript or its supporting information, or deposited to a public repository. For example, in addition to summary statistics, the data points behind means, medians and variance measures should be available. If there are restrictions on publicly sharing data—e.g. participant privacy or use of data from a third party—those must be specified.requires authors to make all data underlying the findings described in their manuscript fully available without restriction, with rare exception (please refer to the Data Availability Statement in the manuscript PDF file). The data should be provided as part of the manuscript or its supporting information, or deposited to a public repository. For example, in addition to summary statistics, the data points behind means, medians and variance measures should be available. If there are restrictions on publicly sharing data—e.g. participant privacy or use of data from a third party—those must be specified.

Reviewer #4: Yes

Reviewer #5: Yes

5. Is the manuscript presented in an intelligible fashion and written in standard English?

Reviewer #4: Yes

Reviewer #5: Yes

6. Review Comments to the Author

Reviewer #4: The authors have adequately addressed all major concerns raised in the previous review. The retrospective nature of the study is now clearly stated, the sensitivity-specificity trade-off is critically discussed, failure analysis has been added, objective texture preservation metrics are provided, and limitations regarding single-device training and blinding are acknowledged. The manuscript is now suitable for publication pending minor corrections.

Minor Comments

1. In Table 6, the note states "Higher CR, CNR value represents better performance," which was copied from Table 5. This is incorrect for FID scores, where lower values indicate better performance. Please correct the note to accurately reflect the metric interpretation.

2. In Table 7, the column header "NEGCUT" appears to be truncated as "NEGCU." Please verify and correct the column header.

3. In the Conclusion section, the phrase "thereby helping clearer visualization" is grammatically awkward. Consider revising to "thereby enabling clearer visualization" or "thereby helping achieve clearer visualization."

4. References 46–49 lack complete bibliographic information (e.g., page numbers). Please ensure these references conform to the journal's citation format requirements.

5. In the Abstract, the phrase "supporting more accurate assessment" may still overstate the findings, given that overall diagnostic accuracy (ROC curves) did not significantly improve. The authors may consider a more conservative phrasing such as "potentially supporting clinical assessment" or "improving image quality that may support clinical interpretation."

Reviewer #5: This manuscript addresses an important clinical problem with a well-designed deep learning approach. The use of Switchable CycleGAN for unpaired medical images is appropriate, and the clinical validation by two experienced radiologists strengthens the findings. However, several methodological clarifications are needed before the clinical rigor can be fully assessed.

The topic is within scope for PLOS ONE. However, the current version has major reporting and methodological gaps, so it is hard to judge the technical rigor and the clinical meaning of the results.

Major comments

1. Regarding the data source, it is unclear if the analysis used raw ultrasound data or images retrieved from the PACS system. If the study used PACS images, the compression levels might differ between the older and newer machines. This could affect the consistency of the training data. Please clarify the technical standards for image storage in the methods section.

2. The choice of a Switchable CycleGAN is appropriate for unpaired medical images. However, the manuscript should explain why this specific architecture is better for preserving clinical features than standard models. Generative models can sometimes create artificial patterns or hallucinations. Please discuss how you ensured that the enhancement process did not alter the actual pathology of the liver.

3. The manuscript presents only successful enhancement cases. To provide a balanced clinical assessment, please include representative failure-case images showing when enhancement is suboptimal or counterproductive, as well as a quantitative analysis of failure patterns (e.g., severe artifacts, obesity, heterogeneous parenchyma). Please also provide clinical guidelines on when not to apply the algorithm. This is critical for safe clinical deployment.

4. The validation set includes 100 cases but only 22 patients are in the non-CLD group. This small sample size for the control group may limit the statistical power of your findings. It is important to prove that the algorithm does not create false cirrhosis-like textures in healthy livers. I suggest adding more control cases or addressing this limitation more clearly. Additionally, for the LC diagnosis comparison (n = 47), please provide post-hoc power analysis for the observed sensitivity improvement (53.2 -> 66.0%, reviewer 1).

5. This study uses only one device pair (12-year-old Aloka -> 4-year-old Canon). The model essentially performs device-to-device translation rather than generalizable quality enhancement. Please discuss expected performance on images from other manufacturers and whether the approach requires retraining for each device pair. Please also address the feasibility of multi-vendor training or transfer learning. This limitation substantially affects the clinical applicability of your method.

6. The comparison with CT or MRI was performed within a six-month window. This time gap is relatively long because liver conditions can progress or change during that period. Please explain the rationale for this timeframe. Also, the manuscript lacks a description of the CT and MRI protocols used as the reference standard.

Minor comments

1. Please add page and line numbers.

2. Consider merging redundant sections (e.g., Introduction vs. Background) and tighten the narrative.

3. Standardize abbreviations and define each once (avoid re-defining common terms).

4. Fix table numbering and cross-references throughout.

5. Check reference numbering using a reference manager.

6. Clean up formatting issues (extra spaces, inconsistent capitalization, inconsistent singular/plural such as “test dataset(s)”).

7. Make figure legends self-contained and ensure they match figure numbering.

8. Please report the inference time per image and discuss the computational requirements for potential real-time deployment during actual scanning procedures.

7. PLOS authors have the option to publish the peer review history of their article (what does this mean?). If published, this will include your full peer review and any attached files.). If published, this will include your full peer review and any attached files.). If published, this will include your full peer review and any attached files.). If published, this will include your full peer review and any attached files.

...

Reviewer #4: No

Reviewer #5: No

---

## [Author Response · Author response to Decision Letter 3]

8 Mar 2026

To the comments of Reviewer 4

1. In Table 6, the note states "Higher CR, CNR value represents better performance," which was copied from Table 5. This is incorrect for FID scores, where lower values indicate better performance. Please correct the note to accurately reflect the metric interpretation.

=> Thank you for the comment. The note has been revised to accurately reflect the interpretation of the metric.

2. In Table 7, the column header "NEGCUT" appears to be truncated as "NEGCU." Please verify and correct the column header.

=> Thank you for the comment. We have revised it.

3. In the Conclusion section, the phrase "thereby helping clearer visualization" is grammatically awkward. Consider revising to "thereby enabling clearer visualization" or "thereby helping achieve clearer visualization."

=>Thank you for the careful comment. We have changed it into “thereby enabling clearer visualization”.

4. References 46–49 lack complete bibliographic information (e.g., page numbers). Please ensure these references conform to the journal's citation format requirements.

=>Thank you for the comment. We have revised it in more detail.

5. In the Abstract, the phrase "supporting more accurate assessment" may still overstate the findings, given that overall diagnostic accuracy (ROC curves) did not significantly improve. The authors may consider a more conservative phrasing such as "potentially supporting clinical assessment" or "improving image quality that may support clinical interpretation."

=> Thank you for the careful comment. We have changed it into “potentially supporting clinical assessment”.

To the comments of Reviewer 5

1. Regarding the data source, it is unclear if the analysis used raw ultrasound data or images retrieved from the PACS system. If the study used PACS images, the compression levels might differ between the older and newer machines. This could affect the consistency of the training data. Please clarify the technical standards for image storage in the methods section.

=>Thank you for highlighting the need to clarify the image source and storage standards. In this retrospective study, all images were obtained from archived clinical ultrasound examinations exported in a consistent manner (not raw RF data). =>We have revised the Dataset section to clearly describe the data source (PACS/exported images), file format and bit depth, and archival compression policy. We also address the potential impact of vendor- or timedependent compression differences and specify the measures taken to minimize bias, including a consistent export pipeline and standardized preprocessing.

2. The choice of a Switchable CycleGAN is appropriate for unpaired medical images. However, the manuscript should explain why this specific architecture is better for preserving clinical features than standard models. Generative models can sometimes create artificial patterns or hallucinations. Please discuss how you ensured that the enhancement

process did not alter the actual pathology of the liver.

=> We thank the reviewer for this important comment. The Switchable CycleGAN uses a single shared generator with AdaIN-based modulation, which constrains bidirectional translation within a unified feature space and reduces the risk of structural drift compared with the conventional two-generator CycleGAN. Cycle-consistency and identity losses further

discourage unnecessary structural changes, while AdaIN mainly adjusts feature statistics rather than generating new spatial structures. To ensure pathology was not altered, we conducte quantitative structure-preservation analyses (patch-wise SSIM and GLCM metrics) These points have been clarified in the Discussion section.

3. The manuscript presents only successful enhancement cases. To provide a balanced clinical assessment, please include representative failure-case images showing when enhancement is suboptimal or counterproductive, as well as a quantitative analysis of failure patterns (e.g., severe artifacts, obesity, heterogeneous parenchyma). Please also provide

clinical guidelines on when not to apply the algorithm. This is critical for safe clinical deployment.

=> Thank you for comments. Representative failure-case images have been added. However, there are limitations to performing a detailed quantitative analysis of failure patterns, as the two reviewers evaluated image quality using a 5-point scale, which is inherently semi-qualitative. For this algorithm, we believe that prioritizing adaptation and modification according to each ultrasound system is necessary before more robust quantitative failure analysis can be conducted.

4. The validation set includes 100 cases but only 22 patients are in the non-CLD group. This small sample size for the control group may limit the statistical power of your findings. It is important to prove that the algorithm does not create false cirrhosis-like textures in healthy livers. I suggest adding more control cases or addressing this limitation more clearly.

Additionally, for the LC diagnosis comparison (n = 47), please provide post-hoc power analysis for the observed sensitivity improvement (53.2 -> 66.0%, reviewer 1).

=>Thank you for your comments. We have described the limitation of the small sample size of the non-CLD group in the Discussion section. In addition, post-hoc power analysis showed that the post-hoc power for the sensitivity improvement observed by Reviewer 1 (53.2 → 66.0%) was approximately 0.28.

5. This study uses only one device pair (12-year-old Aloka -> 4-year-old Canon). The model essentially performs device-to-device translation rather than generalizable quality enhancement. Please discuss expected performance on images from other manufacturers and whether the approach requires retraining for each device pair. Please also address the feasibility of multi-vendor training or transfer learning. This limitation substantially affects the clinical applicability of your method.

=> We appreciate the reviewer’s comment regarding generalizability. We agree that the current model was trained on a single device pair and primarily performs device-to-device translation.Because ultrasound image characteristics vary across vendors, direct deployment to unseen systems may require retraining or fine-tuning. However, as the proposed framework is fully unsupervised and does not require paired data, adaptation to new vendors can be achieved using relatively small unpaired datasets. Multi-vendor training is also feasible through a multidomain translation framework or transfer learning. We have clarified these limitations and potential solutions in the revised Discussion section.

6. The comparison with CT or MRI was performed within a six-month window. This time gap is relatively long because liver conditions can progress or change during that period. Please explain the rationale for this timeframe. Also, the manuscript lacks a description of the CT and MRI protocols used as the reference standard.

=> We thank the reviewer for this important comment. The six-month interval was selected to reflect routine clinical surveillance practice in chronic liver disease, where CT or MRI is typically performed every 3–6 months. Given the retrospective design, strict temporal matching was limited by available data. The evaluated conditions (liver cirrhosis, fatty liver, and focal lesions) are generally chronic structural abnormalities that do not rapidly change, although minor progression cannot be excluded. We have clarified this limitation in the revised manuscript. We have also added detailed descriptions of the CT and MRI acquisition protocols, including contrast phases and diagnostic criteria, in the Methods section.

7. Please add page and line numbers.

=> Thank you for the comment. We have added it.

8. Consider merging redundant sections (e.g., Introduction vs. Background) and tighten the narrative.

=> Thank you for this suggestion. Rather than removing the Background section, we revised the manuscript by reducing redundancy between the Introduction and Background while preserving key references. The Introduction now focuses on clinical motivation, knowledge gaps, and study objectives, whereas the Background has been condensed to provide a concise review of prior work relevant to ultrasound image enhancement and image-to-image translation.

9. Standardize abbreviations and define each once (avoid re-defining common terms).

=> Thank you for this comment. We have revised the manuscript to standardize the use of abbreviations throughout the text.

10. Fix table numbering and cross-references throughout.

=>Thank you for this comment. We have carefully revised the manuscript to standardize table numbering and cross-references throughout the text.

11. Check reference numbering using a reference manager.

=> We thank the reviewer for this comment. We have carefully reviewed all references using a reference management tool and corrected any numbering inconsistencies.

12. Clean up formatting issues (extra spaces, inconsistent capitalization, inconsistent singular/plural such as “test dataset(s)”).

=>Thank you for this suggestion. We carefully revised the manuscript to correct formatting inconsistencies, including extra spaces, capitalization, and singular/plural usage.

13. Make figure legends self-contained and ensure they match figure numbering.

=> We thank the reviewer for this suggestion. We have revised all figure legends to make them fully self-contained. We have also carefully checked and corrected figure numbering to ensure consistency between the text and figure captions.

14. Please report the inference time per image and discuss the computational requirements for potential real-time deployment during actual scanning procedures.

=> Thank you for this important comment. We have added the inference time per image and computational requirements to the Methods section and discussed the feasibility of real-time deployment in the Discussion. The proposed CycleGAN-based model processes a single ultrasound image in approximately 30–40 [ms] on an NVIDIA RTX 3090 GPU, corresponding to 25–30 FPS under single-image inference conditions, suggesting potential applicability for real-time enhancement during clinical ultrasound scanning.

---

## [Decision Letter · Decision Letter 3]

13 Apr 2026

Image quality improvement of liver ultrasound using unsupervised deep learning

PONE-D-23-32222R3

Dear Dr. Lee,

We’re pleased to inform you that your manuscript has been judged scientifically suitable for publication and will be formally accepted for publication once it meets all outstanding technical requirements.

Kind regards,

Do Young Kim, MD, PhD

Academic Editor

PLOS One

Additional Editor Comments (optional):

Reviewers' comments:

Reviewer's Responses to Questions

**Comments to the Author**

1. If the authors have adequately addressed your comments raised in a previous round of review and you feel that this manuscript is now acceptable for publication, you may indicate that here to bypass the “Comments to the Author” section, enter your conflict of interest statement in the “Confidential to Editor” section, and submit your "Accept" recommendation.

Reviewer #4: All comments have been addressed

Reviewer #5: All comments have been addressed

2. Is the manuscript technically sound, and do the data support the conclusions?

Reviewer #4: Yes

Reviewer #5: Yes

3. Has the statistical analysis been performed appropriately and rigorously? 

Reviewer #4: Yes

Reviewer #5: Yes

4. Have the authors made all data underlying the findings in their manuscript fully available?

The PLOS Data policy requires authors to make all data underlying the findings described in their manuscript fully available without restriction, with rare exception (please refer to the Data Availability Statement in the manuscript PDF file). The data should be provided as part of the manuscript or its supporting information, or deposited to a public repository. For example, in addition to summary statistics, the data points behind means, medians and variance measures should be available. If there are restrictions on publicly sharing data—e.g. participant privacy or use of data from a third party—those must be specified.requires authors to make all data underlying the findings described in their manuscript fully available without restriction, with rare exception (please refer to the Data Availability Statement in the manuscript PDF file). The data should be provided as part of the manuscript or its supporting information, or deposited to a public repository. For example, in addition to summary statistics, the data points behind means, medians and variance measures should be available. If there are restrictions on publicly sharing data—e.g. participant privacy or use of data from a third party—those must be specified.requires authors to make all data underlying the findings described in their manuscript fully available without restriction, with rare exception (please refer to the Data Availability Statement in the manuscript PDF file). The data should be provided as part of the manuscript or its supporting information, or deposited to a public repository. For example, in addition to summary statistics, the data points behind means, medians and variance measures should be available. If there are restrictions on publicly sharing data—e.g. participant privacy or use of data from a third party—those must be specified.requires authors to make all data underlying the findings described in their manuscript fully available without restriction, with rare exception (please refer to the Data Availability Statement in the manuscript PDF file). The data should be provided as part of the manuscript or its supporting information, or deposited to a public repository. For example, in addition to summary statistics, the data points behind means, medians and variance measures should be available. If there are restrictions on publicly sharing data—e.g. participant privacy or use of data from a third party—those must be specified.

Reviewer #4: Yes

Reviewer #5: Yes

5. Is the manuscript presented in an intelligible fashion and written in standard English?

Reviewer #4: Yes

Reviewer #5: Yes

6. Review Comments to the Author

Reviewer #4: The authors have satisfactorily addressed all minor comments raised in the previous review. The corrections to table notes, column headers, grammar, reference formatting, and abstract phrasing have been

appropriately implemented.

Reviewer #5: The revised manuscript is improved, and several of my previous concerns have been addressed, including clarification of the image source, additional discussion of the model architecture, inference time, and limitations related to generalizability and reference standards.

However, some concerns remain only partially resolved. In particular, the discussion of failure modes is still not accompanied by sufficiently clear practical guidance on when the algorithm should not be applied or when its outputs should be interpreted with caution. The limitation of the small non-CLD control group is now acknowledged, but this still restricts confidence that cirrhosis-like texture is not being introduced in non-cirrhotic livers. In addition, several typographical and formatting issues remain in the revised manuscript, and the reference list should be checked again for full compliance with PLOS formatting requirements, particularly with respect to DOI presentation and overall consistency.

Overall, the manuscript has improved, but several issues remain insufficiently addressed and should be resolved before the manuscript is considered ready for acceptance.

7. PLOS authors have the option to publish the peer review history of their article (what does this mean?). If published, this will include your full peer review and any attached files.). If published, this will include your full peer review and any attached files.). If published, this will include your full peer review and any attached files.). If published, this will include your full peer review and any attached files.

...

Reviewer #4: No

Reviewer #5: No

---

## [Editor Report · Acceptance letter]

PONE-D-23-32222R3

PLOS One

Dear Dr. Lee,

I'm pleased to inform you that your manuscript has been deemed suitable for publication in PLOS One. Congratulations! Your manuscript is now being handed over to our production team.

Kind regards,

on behalf of

Prof. Do Young Kim

Academic Editor

PLOS One